# Free Phenolic Compounds, Antioxidant Capacity and FT-NIR Survey of Debittered *Lupinus mutabilis* Seeds

**Lorenzo Estivi** [1], **Silvia Grassi** [1], **Luis Briceño-Berrú** [2], **Patricia Glorio-Paulet** [2,3], **Felix Camarena** [4], **Alyssa Hidalgo** [1,*] **and Andrea Brandolini** [5]

1    Department of Food, Environmental and Nutritional Sciences (DeFENS), Università degli Studi di Milano, Via Celoria 2, 20133 Milan, Italy

2    Departamento de Ingeniería de Alimentos y Productos Agropecuarios (DIAPA-FIAL), Facultad de Industrias Alimentarias, Universidad Nacional Agraria La Molina, Av. La Molina s/n, Lima 15024, Peru

3    Instituto de Investigación de Bioquímica y Biología Molecular (IIBBM), Universidad Nacional Agraria La Molina (UNALM), Av. La Molina s/n, Lima 15024, Peru

4    Programa de Leguminosas, Facultad de Agronomia, Universidad Nacional Agraria La Molina (UNALM), Av. La Molina s/n, Lima 15024, Peru

5    Consiglio per la Ricerca in Agricoltura e l'analisi dell'economia Agraria-Unità di Ricerca per la Zootecnia e l'Acquacoltura (CREA-ZA), Via Piacenza 29, 26900 Lodi, Italy

\*    Correspondence: alyssa.hidalgovidal@unimi.it; Tel.: +39-02-5031-9189

**Abstract:** *Lupinus mutabilis* protein-rich seeds must be debittered before consumption. The aim of this research was to assess free phenolic compounds, antioxidant capacity and FT-NIR spectra of flours from debittered seeds of 33 Andean ecotypes of *L. mutabilis,* and five varieties belonging to *L. luteus*, *L. angustifolius* and *L. albus,* as controls. The free phenolics were quantified by RP-HPLC, while the antioxidant capacity was evaluated spectrophotometrically through the Reducing Power, ABTS, FRAP and DPPH methods. The free phenolics of *L. mutabilis* were mostly (85.5–99.6%) flavonoids (genistein and genistein derivatives, apigenin, catechin and naringenin). Other compounds, detected in low quantities, were phenylethanoids (tyrosol and tyrosol derivative) and phenolic acids (cinnamic acid derivatives). The highest total free phenolic concentration was observed in H6 INIA BP (1393.32 mg/kg DM), followed by Chacas, Moteado beige, Huánuco and Lircay. The antioxidant capacity of the *L. mutabilis* ecotypes exceeded that of the controls and was correlated to flavonoids content. Additionally, a relationship between free phenolic compounds and spectral bands was established by FT-NIR, paving the way for a fast, reliable and non-destructive approach to lupin seeds characterisation. Even after debittering, lupin flours maintained high free phenolic concentrations and antioxidant capacity.

**Keywords:** flavonoids; high performance liquid chromatography (HPLC); lupin; phenylethanoids; phenolic acids

## 1. Introduction

The growing interest of consumers for nutritional quality and the beneficial effects of foods on health is constantly stimulating the search for new, raw materials with good macronutrients composition and rich in bioactive compounds. In this context, the rediscovery of traditional but often underutilized crops from different regions of the world is an appealing and realistic alternative. The Andean lupin (*L. mutabilis* Sweet), locally called tarwi or chocho, is a South American legume belonging to a genus which includes three species of worldwide agricultural interest, i.e., *L. albus*, *L. luteus* and *L. angustifolius* [1]. The Andean lupin, cultivated for millennia in the temperate and cold climates of the Andes in South America, combines thrifty agronomic requirements with excellent nutritional properties, thus representing a promising resource for the preparation of high nutritional value food products.

Lupin seeds have a balanced composition, characterised by an excellent content of proteins, lipids and micronutrients [2–4], as well as by numerous bioactive compounds with antioxidant capacity, such as tocols [5–7], carotenoids [6,7] and phenolics [7–9]. Particularly interesting are the phenolic compounds, primary antioxidants that stabilize free radicals protecting cell membranes, proteins, lipids and DNA from oxidative damage [10]. Lupins also contain quinolizidine alkaloids, bitter and toxic compounds which must be removed before consumption, through a long debittering procedure that includes boiling and repeated water washes [11–13]; to avoid this process, some breeding programmes are selecting genotypes characterized by low levels of alkaloids [4,14,15]. Once debittered, lupin seeds have many potential applications as food for humans and animals, and as additives for cosmetic, pharmaceutical and medical industries [15,16].

Most of the published results on the phenolic composition and antioxidant capacity of lupins are about Mediterranean lupins, and refer to bitter seeds. However, the boiling and repeated washings needed for debittering induce a drastic reduction of water-soluble compounds, such as minerals, mono-, di- and oligosaccharides [11,13] and phenolics [8,17,18]. In fact, after debittering, a 96–97% reduction in total polyphenols across three *L. mutabilis* ecotypes was observed [18], while in three other Andean lupins an average of 76% phenolics decrease was noticed; phenylethanoids (−95%) and flavonoids (−71%) were particularly susceptible, while phenolic acids (−57%) seemed to be more resilient [8].

The industry is looking for rapid, low-cost, eco-friendly but accurate methods for chemical composition determination. Among rapid measurement techniques, Near Infrared (NIR) spectroscopy was demonstrated to be a reliable approach for food product characterisation. Indeed, NIR spectra, deriving from the interaction of light-matter in the region of 12,500–400 cm$^{-1}$, contains abundant information on the molecular interaction between functional groups.

The widespread industrial utilisation of the Andean lupin is hampered by the limited number of studies analysing its composition and technological characteristics, as well as the lack of improved varieties suitable for cultivation in the main lupin cropping areas. A recent study characterized chemical composition, carotenoids and tocols of 33 Andean ecotypes of *Lupinus mutabilis,* originating from different ecogeographic regions of Perú [6]. To identify the most suitable ecotypes for developing high nutritional value food products, the aim of this research was to improve the characterization of these very same materials by determining the free phenolic compounds, and antioxidant capacity, of the water-debittered seeds. A second aim was to perform a preliminary investigation of FT-NIR suitability as a fast, reliable and non-destructive approach to assess the antioxidant properties of lupins.

## 2. Materials and Methods

### 2.1. Materials

The seeds of the 33 *Lupinus mutabilis* ecotypes analysed (Supplementary Table S1), kindly provided by the Leguminous Program of the Universidad Nacional Agraria La Molina, Lima, Perú, were originally collected in different regions of the Peruvian Andes (Supplementary Figure S1). One *Lupinus albus* from Perú (cv. Dulce 7), as well as two *L. albus* (cv. Ares and Multitalia), one *Lupinus angustifolius* (cv. Boregine) and one *Lupinus luteus* (cv. Percoz) from Italy were tested as controls (Supplementary Table S1).

### 2.2. Methods

#### 2.2.1. Debittering and Milling

To remove the alkaloids, debittering was conducted according to Córdova-Ramos et al. [11]. The lupin beans were hydrated at room temperature for 12 h with a 1:6 (*w/v*) seeds:water ratio; boiled for 1 h (hydrated seeds:water 1:3 *w/v*), changing water after 30 min; soaked in water (cooked seeds:water 1:3 *w/v*) at room temperature for 5 days, substituting the water every day; strained and then dried at 50 °C in a SW-10S dryer (Xinhang, Henan, China) for 18 h; and stored in the dark at room temperature until milling. This method

efficiently removes alkaloids and bitterness, as demonstrated by Estivi et al. [19], using colorimetric titration and electronic tongue.

The debittered lupin grains were ground with a Grindomix GM 200 knife mill (Retsch GmbH, Germany) at 6000 RPM for 35 s. The whole meals were sifted through a 20.0 mesh (0.85 mm) sieve, vacuum-packed in high-density polyethylene bags and stored at 4 °C until the analysis.

### 2.2.2. Free Phenolic Compounds

The extraction of free phenolic compounds was carried out according to the procedure reported by Brandolini et al. [8]. Approximately 1 g of flour was weighed into capped centrifuge tubes and 7 mL of a methanol solution (80%) was added. After vortexing with a Wizard Advanced IR Vortex Mixer (Velp Scientifica, Usmate, Italy) for about 45–60 s, the samples were placed in an ultrasonic bath for 10 min, vortexed again for a few seconds and centrifuged at 12,000 RPM for 10 min at 8–9 °C with a LISA centrifuge (AFI Groups, France). The supernatants, which contain the free phenolics, were recovered in a 250 mL flask and covered with foil to protect the content from light. The extraction was repeated two more times. The three extracts were merged, evaporated under vacuum using a Laborota 4000 rotavapor (Heidolph, Milan, Italy) for 40 min at 35 °C and completely dried by nitrogen flow for 1 min.

The dry samples were resuspended with 2 mL of methanol:chromatographic water (8:2 *v/v*), vortexed and filtered on a 0.45 μm PTFE membrane (Diana Beck Scientific, Angera, Italy). A 20 μL volume of filtrate underwent reverse phase HPLC analysis [8] using a column Adamas® C18-AQ 5 μm 4.60 0mm × 250 mm and a precolumn C18 5 μm 4.60 mm × 10.0 mm (SepaChrom SRL, Rho, Italy) thermostated at 30.0 °C; L-2130 pump, L-2300 column oven and L2450 Diode Array Detector Elite LaChrom (Hitachi, Tokyo, Japan).

The identity of the compound was confirmed by the congruence of retention times and UV/VIS spectra, with those of pure authentic standards. The unidentified peaks were quantified using the calibration curve of the compound with a similar absorption spectrum and named as "phenolic derivative", as indicated in Brandolini et al. [8], and according to Dueñas et al. [20] and Zalewski et al. [21]; additionally, the derivatives of each compound were grouped together. For phenolics quantification, the calibration curves of the identified phenolics were constructed using standards (Sigma-Aldrich) recorded at 280 nm for catechin (13.9–99.2 mg/L), genistein (27.5–110 mg/L), naringenin (2.25–9.00 mg/L), tyrosol (3.93–98.2 mg/L), cinnamic acid (4.05–19.1 mg/L), at 320 nm for apigenin (1.00–20.0 mg/L) and at 360 nm for diosmin (5.24–104.80 mg/L). Based on the calibration curve, the limit of detection was calculated as the intercept value of the regression line, plus three times the standard error of the estimate [22]. The calibration curves were linear in the concentration intervals assessed, with the respectively following detection limits: 1.86, 1.52, 0.15, 1.40, 0.19, 1.19 and 0.41 mg/L. The results are reported as mg/kg DM.

### 2.2.3. Antioxidant Capacity

Exactly $0.100 \pm 0.010$ g of flour was mixed in an Eppendorf tube with 1 mL of 80% methanol solution, vortexed, placed in an ultrasonic bath for 10 min, shaken for 20 min and centrifuged at 12,000 rpm for 10 min. After recovering the supernatant, the extraction was repeated and the two supernatants combined. The antioxidant capacity of debittered lupins was evaluated by the reducing power (RP) method, according to Oyaizu [23], the ABTS method according to Re et al. [24], the FRAP method and the DPPH method according to Yilmaz, Brandolini and Hidalgo [25]. The antioxidant capacity was expressed as millimoles Trolox equivalent (TE)/kg dry matter (DM).

All the assays were carried out in three independent samples.

### 2.2.4. Near Infrared Spectroscopy

The debittered lupin flours were analysed by a Fourier Transform (FT)-NIR spectrometer (MPA, Bruker Optics, Ettlingen, Germany) equipped with an integrating sphere module,

with a 25 mm diameter window, working in diffuse reflectance mode and a RT-PbS external detector (non-linearity correction: cut-off 3350 cm$^{-1}$, efficiency: 0.9). Two 1 g aliquots of each sample were analysed by positioning the glass cups containing the sample on the measuring window. The spectra were collected in the range 12,500 to 3600 cm$^{-1}$ (resolution, 8 cm$^{-1}$; scanner velocity, 10 kHz; background, 32 scans; sample, 32 scans). OPUS software (v. 6.5, Bruker Optics, Ettlingen, Germany) was used for instrumental control and for spectra acquisition.

2.2.5. Statistical Analysis

To assess the differences among Andean lupins, species or geographical area of origin for the traits analysed, a one-way analysis of variance (ANOVA) was performed. The geographical areas were: North (groups A + B), Centre North (C + D), Centre South (E + F) and South (G + H). Normal distribution of the data was verified and, if needed, it was log-transformed. When significant differences were detected, Fisher's least significant differences (LSD) at $p \leq 0.05$ were calculated. Pearson's linear correlation between the total free phenolic acid content of lupins, and their antioxidant capacity, was also determined. All these analyses were performed using the STATGRAPHICS® Centurion statistical programme (StatPoint Technologies Inc., Warrengton, VA, USA). The average values, standard deviations and standard errors were computed using the Excel® program (Microsoft, Redmond, WA, USA).

Furthermore, Principal Component Analysis (PCA) was applied to explore sample distribution of debittered seeds, according to species or geographical area of origin, for the traits analysed on the same samples in this work (free phenolics and antioxidant capacities) and in previous related research (carotenoids and tocols) [6], as well as to assess variable relationships [26]. Regression models were developed by partial least squares (PLS), combining NIR spectral information (X) with the dependent information (Y), i.e., phenolic compounds and antioxidant capacity data. The spectral range was reduced to the most informative region (9000–3800 cm$^{-1}$) and transformed by different spectral pre-processing techniques (alone or in combination), i.e., no spectral pre-processing; standard normal variation (SNV); smoothing (Savitzky-Golay with filter width 15); first derivative (Savitzky-Golay with filter width 15); second derivative (Savitzky-Golay with filter width 15) and always followed by mean centring. In addition, a variable selection procedure was performed using interval Partial Least Square (i-PLS) [27], to identify the best subset of spectral variables for the elaboration of the PLS regression models. The model dimensionality, i.e., the number of latent variables (LV), was assessed by the venetian blind cross-validation strategy. Statistical parameters such as root mean error in calibration (RMSE$_{CAL}$) and cross-validation (RMSE$_{CV}$); root mean square percentage error in calibration (RMSPE$_{CAL}$) and cross-validation (RMSPE$_{CV}$); coefficient of determination of both calibration ($R^2_{CAL}$) and cross-validation ($R^2_{CV}$) and bias and ratio of performance to deviation (RPD) were used to assess model performance. The PCA and PLS models were developed with PLS Toolbox (Eigenvector, Manson, WA, USA) working under MATLAB environment (R2016b, The Mathworks, Natick, MA, USA).

## 3. Results

### 3.1. Phenolics Composition and Content

The phenolic composition of the analysed debittered Andean lupins is reported in Supplementary Table S2. Nine compounds were detected: the flavonoids genistein; genistein derivative; apigenin derivative; catechin derivative; diosmetin derivative; naringenin derivative; phenylethanoids tyrosol; tyrosol derivative and the phenolic cinnamic acid derivative. The most abundant phenolics were the flavonoids, encompassing 85.3–99.6% of total phenolics, followed by the phenylethanoids (not detected-13.5%) and by the phenolic acids (not detected-2.1%). Genistein derivative (464.69 ± 25.30 mg/kg DM) and genistein (135.78 ± 10.49 mg/kg DM) represented 67.4% and 19.7% of total flavonoids, respectively, corresponding to 63.6% and 18.6% of total phenolics. Four molecules (genistein, genistein

derivative, apigenin and diosmetin) were present across all Andean lupins, while others were below the detection limits in one (naringenin derivative) or more (catechin derivative, tyrosol, tyrosol derivative and cinnamic acid derivative) samples. The phenolics profile and concentration was like that described by Brandolini et al. [8] for three debittered Andean lupin ecotypes, but lacking the vanillic acid, *p*-hydroxybenzoic acid and 2,4-hydroxybenzoic acid derivatives.

The ANOVAs (not shown) detected highly significant differences ($p \leq 0.001$) among ecotypes for all the traits analysed. The H6 INIA BP showed the highest total free phenolics concentration (1393.32 mg/kg DM), followed by Chacas (1342.60 mg/kg DM), Moteado beige (1271.02 mg/kg DM), Huánuco 2 (1260.06 mg/kg DM) and Lircay (1221.26 mg/kg DM), while the ecotypes CD Junín 7-2 (419.97 mg/kg DM) and Churibamba (340.81 mg/kg DM) had the lowest (Figure 1). Gálvez-Ranilla et al. [28], studying the isoflavones of bitter seeds from six Andean lupins from the Legume and Cereal program of the Universidad Agraria of Lima, Perú, observed a broad variation, with one sample (H6, the source of our two H6 INIA ecotypes) having isoflavones and antioxidant capacity (DPPH method) far superior to the other *L. mutabilis*, including three accessions present in our research, SCG 22, Yunguyo and Andares (sic; probably a transcription error for Andenes).

The ANOVAs showed the existence of highly significant differences ($p \leq 0.05$ to $p \leq 0.001$) between Andean lupins and the controls for all the phenolic compounds, except apigenin; hence, the differences were highly significant ($p \leq 0.001$) for total phenolics content, too. The lupin controls belonging to the *L. albus* and *L. angustifolius* species had very low phenolics content (6.8–31.3 mg/kg DM) and completely different composition (Table 1), limited to apigenin derivatives and, for Dulce 7, to genistein and its derivative. On the other hand, the *L. luteus* control showed a phenolics content (448.1 mg/kg DM) in the lower range of the *L. mutabilis* tested and a comparable composition, rich in flavonoids. Several authors [29–32] observed that non-debittered *L. luteus* seeds have higher phenolic acids and flavonoids content than *L. albus* and *L. angustifolius*. A higher total phenolic content in *L. mutabilis* is also shown from the results and the principal component analysis performed by Czubinski et al. [7].

In non-debittered seed samples, Gálvez-Ranilla et al. [28] noticed that their *L. albus* and *L. angustifolius* controls had very low or non-detectable concentrations of isoflavones, while apigenin and two *p*-coumaric derivatives were observed in *L. albus* [33], and ferulic, sinapic and *p*-coumaric acids were recorded in *L. angustifolius* [34]. On the other hand, from bitter seeds of the three Mediterranean lupin species, Lampart-Szczapa et al. [35] recovered protocatechuic, *p*-hydroxybenzoic, chlorogenic, vanillic, *p*-coumaric and ferulic acids, while Siger et al. [32] identified apigenin along with gallic, protocatechuic, *p*-hydroxybenzoic, caffeic and *p*-coumaric acids. Further examples of the great intra- and inter-species heterogeneity come from Czubinski et al. [29], who found only apigenin derivatives in the three species *L. albus*, *L. angustifolius* and *L. luteus* (382.4–569.7, 784–1027.3 and 1380.2–1417.7 g/kg DM, respectively), and from Kalogeropoulos et al. [36] who, conversely, reported the prevalence of phenolic acids on flavonoids in *L. albus* (14.1 and 8.2 mg/kg DM, respectively).

The broad differences observed between species (but also within species) were correctly attributed to the effects of genetic selection and long-term adaptation towards eco-geographical areas with different edaphic, climatic and pathogenic characteristics [30]. Additionally, debittering is often unavoidable for sweet varieties as well [18], and this process adds an additional source of variability due to different methods and conditions.

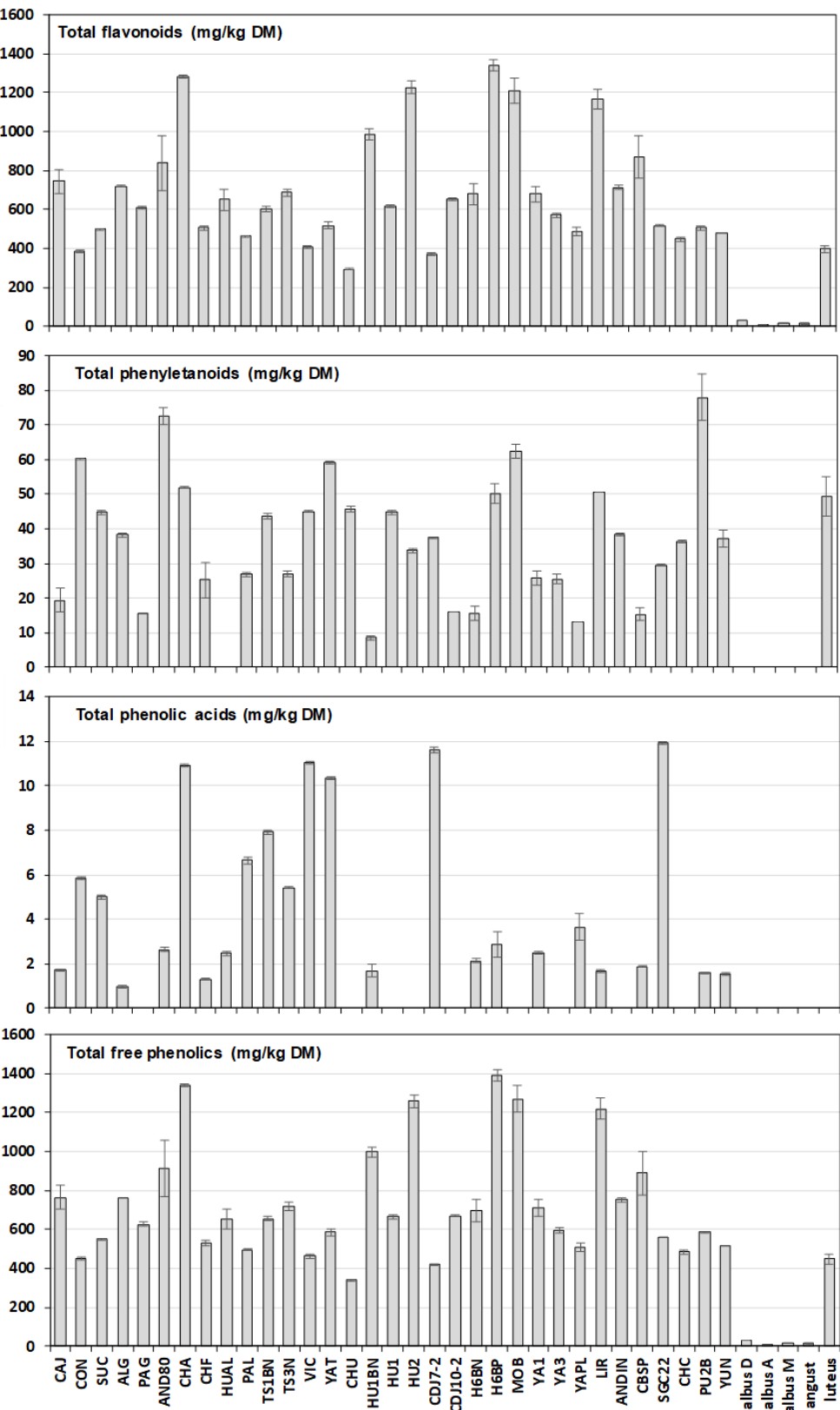

**Figure 1.** Total flavonoids, total phenylethanoids, total phenolic acids and total free phenolics of debittered seeds of 33 *L. mutabilis*, three *L. albus*, one *L. angustifolius* and one *L. luteus*. The ecotype codes are defined in Supplementary Table S1. Error bars represent standard deviation.

**Table 1.** Mean value (±standard error) of free phenolics content (mg/kg DM) and antioxidant capacity (mmol TE/kg DM) of debittered seeds of 33 *L. mutabilis*, three *L. albus*, one *L. angustifolius* and one *L. luteus*. In parentheses, the ranges of the values observed are reported.

| | *L. mutabilis* | *L. albus* | *L. angustifolius* | *L. luteus* |
|---|---|---|---|---|
| | **Free phenolics content** | | | |
| Genistein | 135.78 ± 10.49 (*38.03–352.12*) | 2.21 ± 0.00 (*nd–13.26*) | nd | 40.39 ± 3.89 |
| Genistein der * | 464.69 ± 25.30 (*219.57–1062.84*) | 0.82 ± 0.00 (*nd–4.95*) | nd | 230.00 ± 13.68 |
| Apigenin der | 13.02 ± 1.01 (*2.70–27.99*) | 6.19 ± 0.79 (*6.79–17.31*) | 14.82 ± 1.47 | 27.82 ± 0.39 |
| Catechin der | 6.11 ± 1.06 (*nd–36.49*) | nd | nd | nd |
| Diosmetin der | 64.69 ± 4.35 (*7.23–168.19*) | nd | nd | 98.43 ± 5.47 |
| Naringenin der | 5.36 ± 0.31 (*nd–11.58*) | nd | nd | 2.03 ± 0.07 |
| Tyrosol | 11.30 ± 1.58 (*nd–49.21*) | nd | nd | nd |
| Tyrosol der | 24.90 ± 2.31 (*nd–60.27*) | nd | nd | 49.42 ± 4.07 |
| Cinnamic acid der | 3.49 ± 0.47 (*nd–11.92*) | nd | nd | nd |
| Total free phenolics | 729.36 ± 34.74 (*340.81–1393.32*) | 9.23 ± 0.79 (*6.79–31.27*) | 14.82 ± 1.47 | 448.09 ± 15.84 |
| | **Antioxidant capacity** | | | |
| Reducing power | 2.53 ± 0.15 (*1.09–6.68*) | 1.03 ± 0.19 (*1.44–3.10*) | 0.89 ± 0.02 | 4.01 ± 0.53 |
| ABTS | 40.17 ± 1.74 (*17.25–74.27*) | 7.41 ± 3.03 (*2.36–21.98*) | 9.23 ± 0.49 | 31.34 ± 0.80 |
| DPPH | 2.63 ± 0.14 (*1.06–6.47*) | 0.34 ± 0.06 (*0.25–1.40*) | 0.50 ± 0.02 | 0.86 ± 0.07 |
| FRAP | 18.41 ± 0.74 (*9.48–30.90*) | 2.30 ± 0.06 (*3.90–5.21*) | 5.08 ± 0.11 | 12.44 ± 0.04 |

* der, derivative; nd, not detected, i.e., below the detection limit.

### 3.2. Antioxidant Capacity and Its Correlation with Phenolics Content

The antioxidant capacity of the *L. mutabilis* ecotypes, measured by reducing power, DPPH, ABTS and FRAP, is presented in Figure 2. The ANOVAs (not reported) highlighted the presence of significant differences ($p \leq 0.001$) among samples for all methods. Andenes 80, H6 INIA BN, H6 INIA BP, Moteado beige and Lircay were the ecotypes with the highest values across most tests; conversely, Congona and Churibamba were the ones with the lowest results across most test.

The ABTS average result for *L. mutabilis* (40.2 mmol TE/kg DM) is in the same order of magnitude of non-debittered *L. albus* and *L. angustifolius* analysed by Martínez-Villaluenga et al. [37] (47.2–71.4 mmol TE/kg DM); some ecotypes (e.g., Andenes 80, Puno 2 blanquita, Andenes INIA and Lircay) reached the highest values, 1.5–1.8-fold the mean.

The antioxidant capacity of the controls was significantly different (ANOVA not shown) and inferior to that of the Andean lupins in three out of four tests (DPPH, ABTS and FRAP). Among the European lupins, *L. luteus* showed the highest values (Table 1). Interestingly, the debittered Andean lupin results were similar or only slightly inferior to those recorded in bitter beans of the Mediterranean lupins. In fact, ABTS and FRAP antioxidant capacities of 53–123 µmol TE/g DM and 9.4–12.4 µmol $Fe^{2+}$/g DM were recorded in *L. albus* [33]; 87–152 µmol TE/g extract (ABTS) and 0.108–0.235 µmol $Fe^{2+}$/g extract (RP) were observed in *L. angustifolius* [34]; and DPPH values of 3.51–6.58 µmol TE/g DM in *L. albus*, 6.89–7.54 in *L. angustifolius* and 8.12–9.03 in *L. luteus* were detected [32]. Generally, a decrease in antioxidant activity after debittering is evident, as documented by

several authors [17,18,38], and is most likely due to soluble phenolics thermal degradation and washout [18,39]. In fact, the bound fraction and the lipophilic antioxidants appear unaffected, or even concentrated, by removal of other components [6,8], while quinolizidine alkaloids do not exhibit antioxidant activities [40–42].

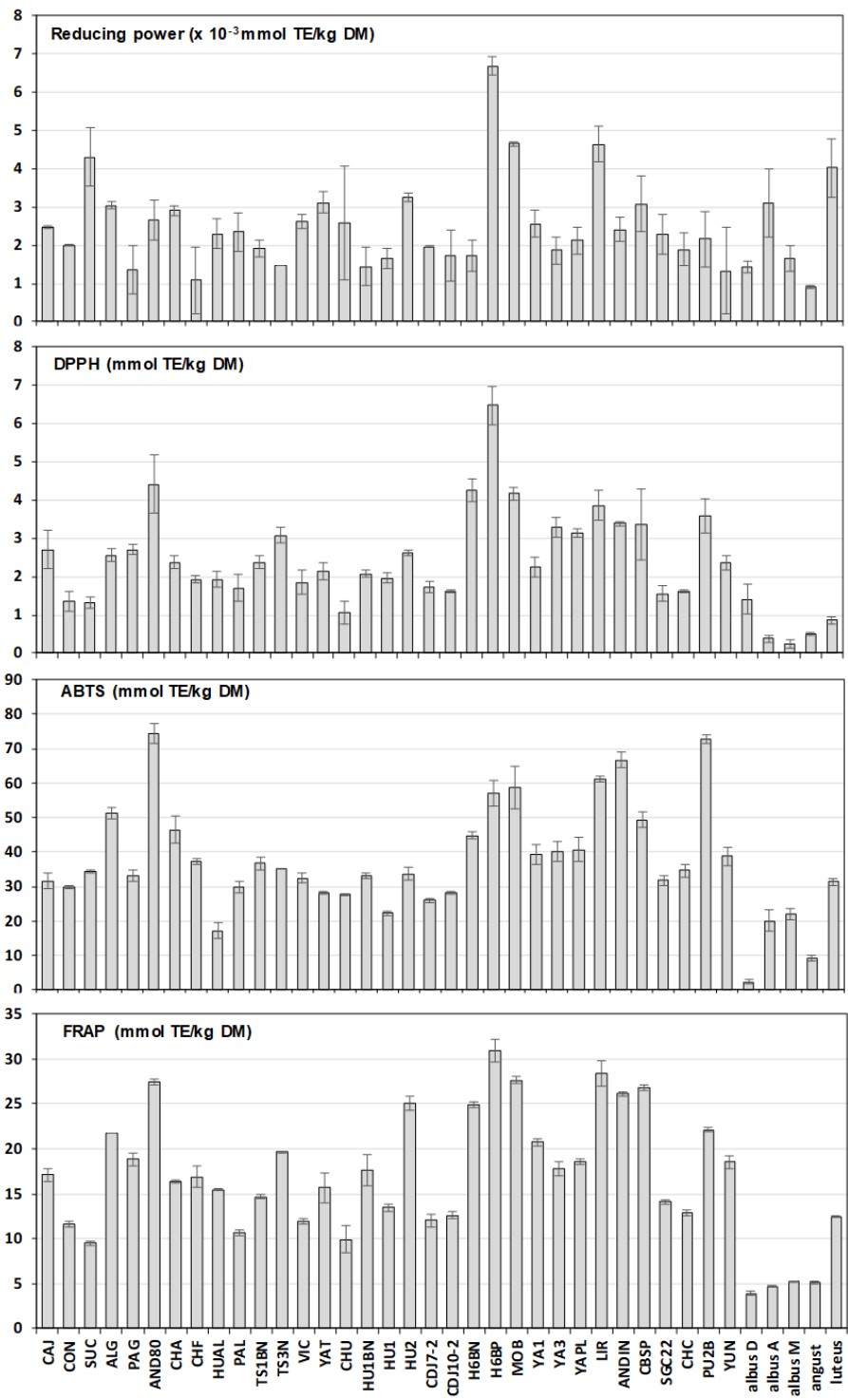

**Figure 2.** Antioxidant capacity (reducing power, DPPH, ABTS and FRAP methods) of debittered seeds of 33 *L. mutabilis*, three *L. albus*, one *L. angustifolius* and one *L. luteus*. The ecotype codes are defined in Supplementary Table S1. Error bars represent standard deviation.

The linear correlations between antioxidant capacity and individual phenolics (Table 2) were mostly significant, with the exceptions of diosmetin derivative, tyrosol derivative and cinnamic acid derivative, which did not present any correlation. The results of the four antioxidant capacity tests were always significantly correlated ($p \leq 0.01$ and $p \leq 0.001$) to flavonoids and total phenolics concentrations, while phenylethanoids and phenolic acids, scarcely present in the debittered Andean lupins, did not show significant values (except phenylethanoids with ABTS). Hassine et al. [41] found significant, very high, Pearson's correlation coefficients between ABTS and total phenolics or phenolic acids (0.948 and 0.905, respectively), although they reported that only tyrosol was correlated with DPPH and FRAP assays.

**Table 2.** Pearson's linear correlation coefficients (n = 38) among antioxidant capacity (RP, reducing power; ABTS, 2,2'-azino-bis (3-ethylbenzothiazoline-6-sulfonic acid); DPPH, 2,2-diphenyl-1-picrylhydrazyl; FRAP, ferric reducing antioxidant power) and free phenolics content of all *Lupinus* samples.

| | RP | ABTS | DPPH | FRAP |
|---|---|---|---|---|
| Genistein | 0.35 * | 0.52 *** | 0.55 *** | 0.69 *** |
| Genistein der | 0.59 *** | 0.60 *** | 0.76 *** | 0.80 *** |
| Apigenin der | 0.38 * | 0.49 ** | 0.47 ** | 0.58 *** |
| Catechin der | 0.05 | 0.42 ** | 0.47 ** | 0.45 ** |
| Diosmetin der | 0.16 | 0.21 | 0.17 | 0.32 |
| Naringenin der | 0.10 | 0.45 ** | 0.34 * | 0.35 * |
| Tyrosol | 0.29 | 0.73 *** | 0.68 *** | 0.74 *** |
| Tyrosol der | 0.29 | 0.20 | −0.02 | −0.04 |
| Cinnamic acid der | 0.04 | −0.04 | −0.05 | −0.11 |
| Flavonoids | 0.61 *** | 0.45 ** | 0.63 *** | 0.71 *** |
| Phenylethanoids | 0.40 | 0.52 *** | 0.22 | 0.15 |
| Phenolic acids | 0.01 | −0.25 | −0.27 | −0.40 * |
| Total phenolics | 0.63*** | 0.48** | 0.64*** | 0.71*** |

*, ** and ***: significant at $p \leq 0.05$, 0.01 and 0.001, respectively; der: derivative.

The results of the PCA performed on the phenolic contents (flavonoids, phenylethanoids, phenolic acids, tocols, carotenoids and total free phenolics) and the antioxidant capacity (reducing power, DPPH, ABTS and FRAP) are shown in Figure 3. The biplot confirmed that *L. albus* and *L. angustifolius* differ from the Andean lupins for all of the phenolic compounds and the antioxidant capacity, whereas *L. luteus* showed phenolic contents and antioxidant capacity comparable to those of *L. mutabilis*. Indeed, the *L. albus* and *L. angustifolius* controls assumed highly negative PC1 scores (<0.02), thus positioning far away from the other samples. On the opposite side of the PC1 axis it was possible to notice the positioning of the ecotypes with the highest total free phenolics concentration, i.e., H6 INIA BP, Lircay, Moteado beige, Chacas, Huánuco 2 and Andenes 80. This result is linked to a higher carotenoids content in *L. albus* and *L. angustifolius* coupled to lower flavonoids, phenylethanoids and phenolic acids contents, as also confirmed by the negative loadings for carotenoids and positive loadings for the other compounds.

The variable loadings confirmed the existing relation between antioxidant capacity and phenolics, as tocols, flavonoids and total phenolics loadings were near DPPH, ABTS and FRAP, indicating that they are related and contribute similarly to sample distribution.

### 3.3. Geographical Origin

The ANOVAs conducted after grouping the Andean lupins in geographical areas, viz. North, Centre-North, Centre-South and South, did not highlight any significant difference for phenolic content among ecotypes of the different regions. This result was corroborated by the PCA (Figure 3), as no sample grouping was observed according to the origin. On the other hand, ABTS, DPPH and FRAP antioxidant capacities were significantly different between groups at $p \leq 0.05$, and RP was significant at $p \leq 0.089$. On average, the ecotypes

coming from the Centre-South and South of Perú showed higher antioxidant capacity than those from the Centre-North and North areas (Table 3).

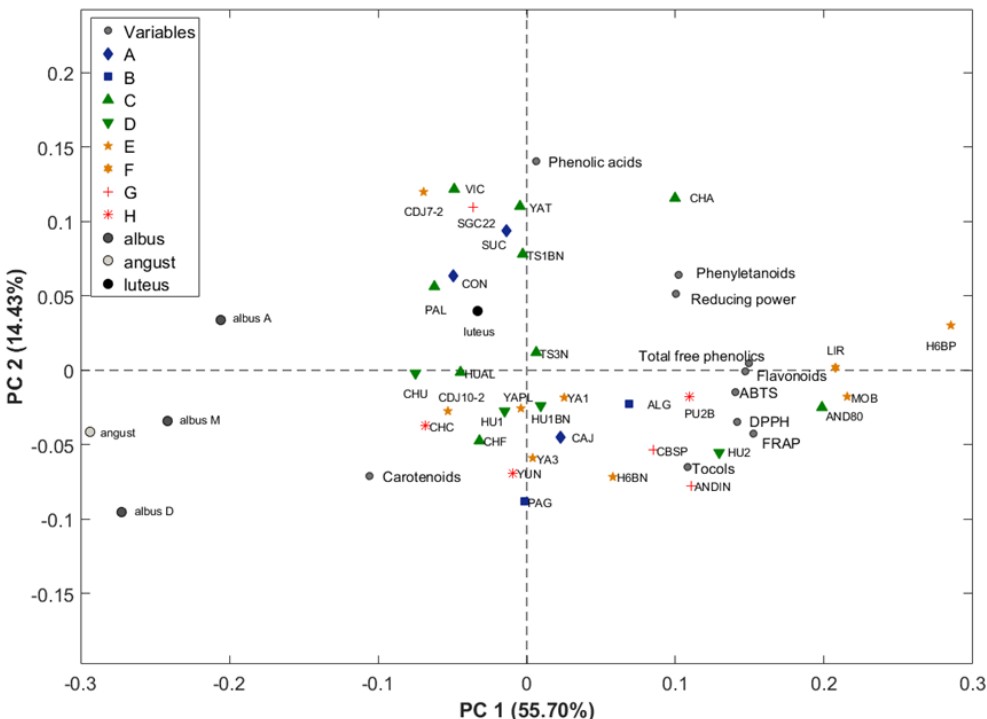

**Figure 3.** Biplot of the principal component analysis performed on phenolic contents (flavonoids, phenylethanoids, phenolic acids, tocols, carotenoids and total free phenolics) and antioxidant capacity (reducing power, DPPH, ABTS and FRAP) of debittered seeds of 33 *L. mutabilis*, three *L. albus*, one *L. angustifolius* and one *L. luteus*. The ecotype codes are defined in Supplementary Table S1. Carotenoids and tocols data are from a previous study [6].

**Table 3.** Mean value (±standard error) of free phenolics (mg/kg DM) and antioxidant capacity (mmol TE/kg DM) of 33 *L. mutabilis* from different regions of Perú. RP, reducing power; ABTS, 2,2'-azino-bis (3-ethylbenzothiazoline-6-sulfonic acid); DPPH, 2,2-diphenyl-1-picrylhydrazyl; FRAP, ferric reducing antioxidant power.

| | Area | | | |
|---|---|---|---|---|
| | **North** | **Centre-North** | **Centre-South** | **South** |
| **N° Ecotypes** | **5** | **13** | **9** | **6** |
| Flavonoids | 592.34 ± 67.65 | 699.20 ± 84.36 | 796.46 ± 116.43 | 589.92 ± 167.02 |
| Phenylethanoids | 35.68 ± 8.27 | 37.25 ± 5.51 | 32.99 ± 5.99 | 39.18 ± 20.89 |
| Phenolic acids | 2.70 ± 1.15 | 4.65 ± 1.19 | 2.72 ± 1.20 | 2.82 ± 4.54 |
| Total phenolics | 630.72 ± 60.76 | 741.10 ± 84.46 | 832.16 ± 120.66 | 631.91 ± 156.70 |
| RP | 2.63 ± 0.50 [ab] | 2.26 ± 0.19 [b] | 3.10 ± 0.59 [a] | 2.19 ± 0.58 [b] |
| ABTS | 36.17 ± 3.90 [bc] | 34.95 ± 3.84 [c] | 44.02 ± 4.24 [ab] | 49.02 ± 17.27 [a] |
| DPPH | 2.13 ± 0.32 [b] | 2.27 ± 0.22 [b] | 3.42 ± 0.50 [a] | 2.64 ± 0.92 [ab] |
| FRAP | 15.73 ± 2.27 [b] | 16.52 ± 1.42 [b] | 21.52 ± 2.29 [a] | 20.09 ± 5.93 [ab] |

Values in the same row with different letters are significantly different ($p \leq 0.05$) following Fisher's LSD test.

Since no geographical diversities were found for polyphenols content, these differences may be due to other antioxidant compounds present in Andean lupins, such as tocols, carotenoids, etc. In fact, Briceño-Berru et al. [6], in these very same ecotypes, found greater concentration of β-tocopherol in samples coming from the Centre-South and South regions of Perú, as well as of lutein and total carotenoids in samples from the South, which were attributed to the selection pressure exerted by the harsher climatic conditions of those environments.

### 3.4. Near Infrared Spectra and Regression Models

All the species showed similar FT-NIR spectra before (Figure 4A) and after the SNV transformation (Figure 4B). An absorption band around 8550–8300 cm$^{-1}$ could be attributed to the second overtone of -CH$_2$, and the relatively wide peak at 6900 cm$^{-1}$ can be assigned to the O–H stretch first overtone of hydroxyl groups [43]. Around 5785 cm$^{-1}$ and 5680 cm$^{-1}$ were the present signals related to the first overtone of C–H stretching [44], while the peak near 5200 cm$^{-1}$ could be attributed to O–H bending/stretching of polysaccharides, converging with those of water; the absorption at 4332 cm$^{-1}$ may be linked to (C–H) bending [43]. The only remarkable difference among the FT-NIR spectra of the different species was in the 4800–4600 cm$^{-1}$ region, where sharp peaks at 4850 cm$^{-1}$ and 4660 cm$^{-1}$ were observed for the Andean lupins and *L. luteus*. This region was previously recognized as the most relevant for the assessment of phenolic contents in faba beans [45], and in tannins and flavonoids [46] due to the asymmetric C–O–O stretches [47]. Hence, the differences detected in this region confirmed the lower total phenolics content of *L. albus* and *L. angustifolius*.

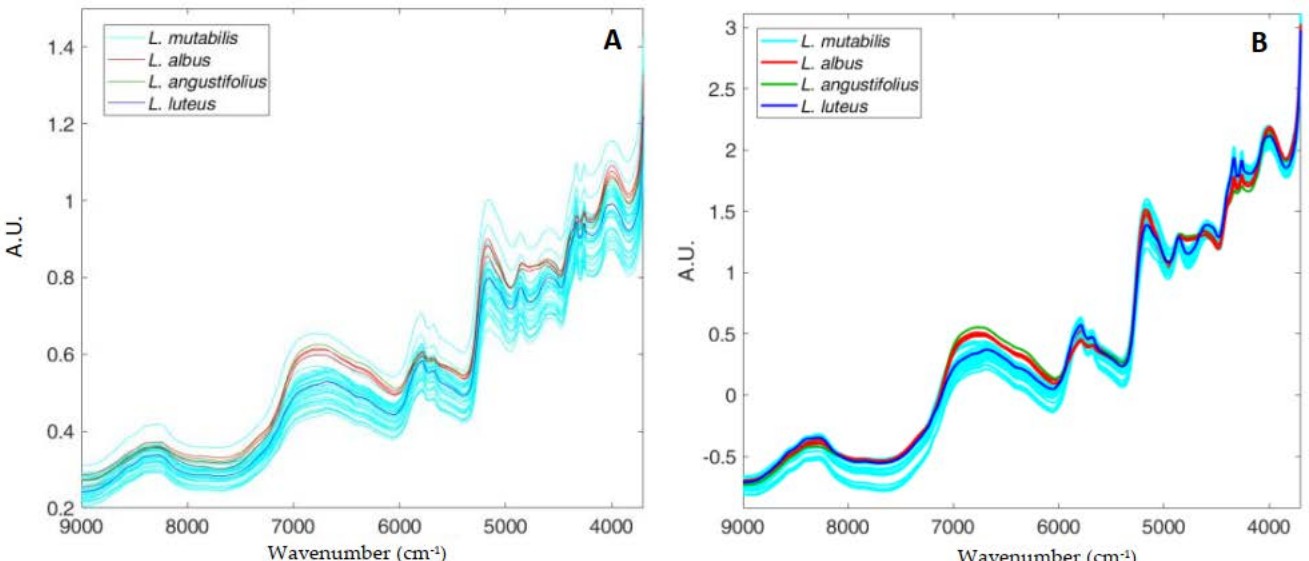

**Figure 4.** FT-NIR spectra collected for the 33 *L. mutabilis* ecotypes, three *L. albus* (cv. Dulce 7, Ares and Multitalia), one *L. angustifolius* (cv. Boregine) and one *L. luteus* (cv. Percoz): (**A**) raw spectra; (**B**) FT-NIR spectra after the SNV transformation.

The PCA performed with the FT-NIR spectra again split the *L. albus* and *L. angustifolius* samples from the other lupins along the PC1 (Figure 5A), linked to PC1 positive loadings of the region 4850 cm$^{-1}$ and 4660 cm$^{-1}$ (Figure 5B), where sharp peaks were observed in the FT-NIR spectra collected for Andean lupins and *L. luteus*. Their differentiation from the other lupins is confirmed by their high total carotenoid values, coupled to low tocols, flavonoids, tyrosol, total phenolic acids contents, antioxidant capacities and *b*\* colour coordinate. Similarly, four Andean lupins (Andenes INIA, Lircay, Altagracia and Puno 2 blanquita) formed a small cluster in the top right quadrant (Figure 5A); their location is linked to high positive loadings, for both PC1 and PC2, in the region 5000–5500 cm$^{-1}$ (Figure 5B), characterised by a peak near 5200 cm$^{-1}$ related to O–H bending/stretching of polysaccharides, converging with those of water. The four ecotypes were always among the top scorers for total tocols content, ABTS and FRAP antioxidant capacity of the methanolic extracts, tyrosol and total free phenolics content, and *b*\* colour coordinate. Finally, the isolated Huallanca ecotype (Figure 5A, in the bottom left quadrant) was characterised by the lowest total carotenoids, tyrosol content and ABTS antioxidant capacity of the methanolic extracts.

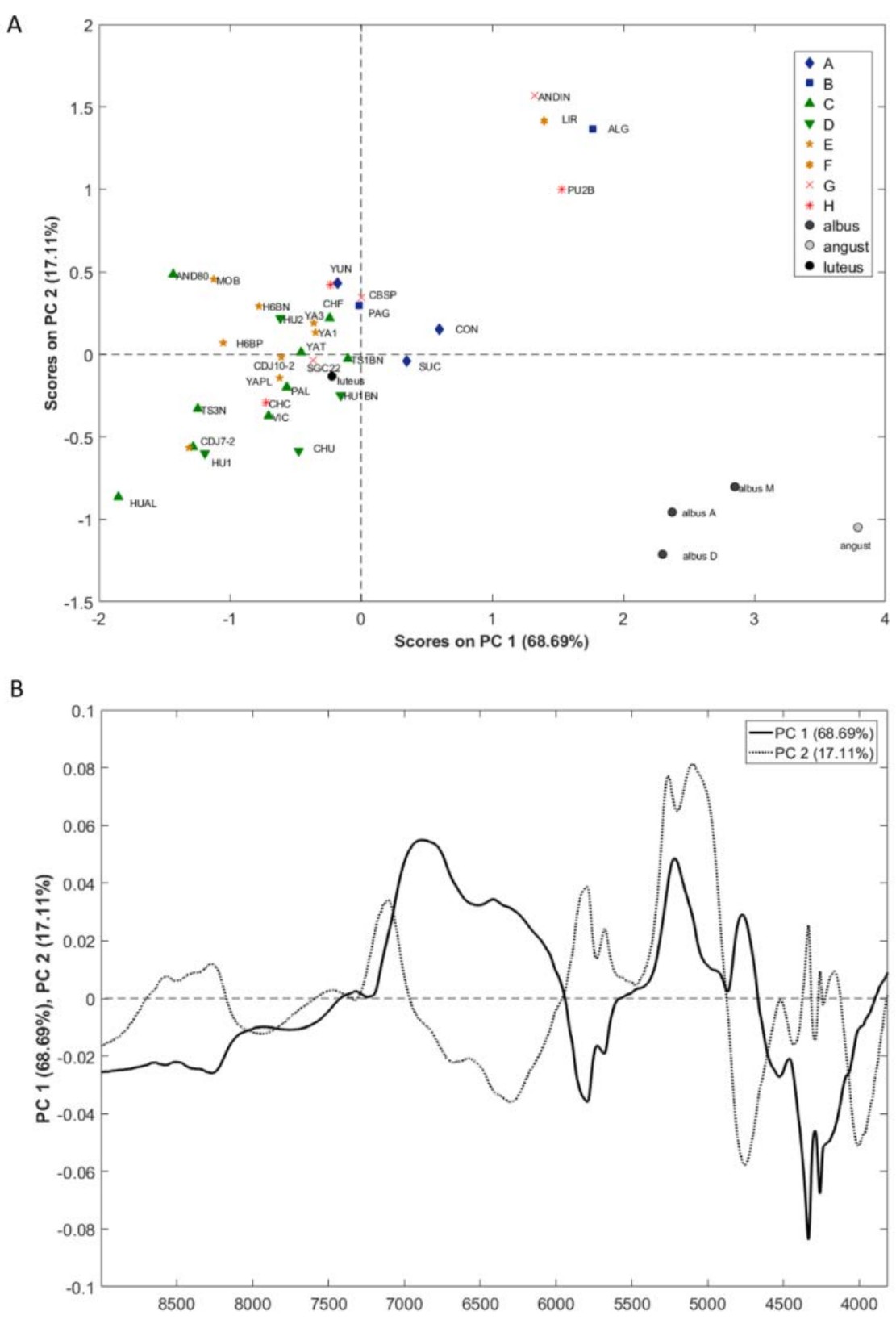

**Figure 5.** Principal component analysis performed on the FT-NIR spectra of debittered seeds of 33 *L. mutabilis*, three *L. albus*, one *L. angustifolius* and one *L. luteus*; the ecotype codes are defined in Supplementary Table S1. (**A**) PC1 vs PC2 scores plot. (**B**) PC1 and PC2 loadings plot.

The relationship between phenolic content and FT-NIR spectra was further investigated by the development of i-PLS regression models. The best models were obtained after SNV transformation for total phenolics ($R^2_{CV}$ of 0.84, $RMSE_{CV}$ of 112.40 mg/kg DM), flavonoids ($R^2_{CV}$ of 0.83, $RMSE_{CV}$ of 116.80 mg/kg DM), carotenoids ($R^2_{CV}$ of 0.86, $RMSE_{CV}$ of 0.51 mg/kg DM), tocols ($R^2_{CV}$ of 0.86, $RMSE_{CV}$ of 23.80 mg/kg DM) and antioxidant capacity as FRAP ($R^2_{CV}$ of 0.76, $RMSE_{CV}$ of 2.89 mmol TE/kg DM); more details are reported in Table 4. The spectral regions selected by the i-PLS algorithm confirmed the relevance of the absorption bands related to phenolics presence. Similar results are reported by Tomar et al. [43] for the prediction of phenolics (range 0.04–0.21 g/100 g) in pearl millet (*Pennisetum glaucum* L.), with standard error in cross validation of 0.0199 g/100 g. Carbas et al. [48] reached higher performance in predicting phenolics and flavonoids (RPD of 5.20 and 5.18, respectively), but poorer performance for FRAP prediction (RPD of 2.02) in comparison to our models. The regression models developed for total phenolics and flavonoids, however, have a range of variability not comparable to ours, impeding any comparison of the results, together with a lower number of samples used for calibration (n = 21).

**Table 4.** Prediction performance of PLS regression models.

| Parameter | Range | Variables (cm$^{-1}$) | LVs | CAL | | | | CV | | | | |
|---|---|---|---|---|---|---|---|---|---|---|---|---|
| | | | | $R^2$ | RMSE | RMSPE | Bias | $R^2$ | RMSE | RMSPE | Bias | RPD |
| Total phenolics (mg/kg DM) | 300.00–1393.32 | 6823–6711, 6245–6133, 4509–4397, 4278–4050 | 7 | 0.92 | 78.50 | 12.1 | $4.55^{-12}$ | 0.84 | 112.40 | 17.4 | $1.11^{-2}$ | 3.0 |
| Flavonoids (mg/kg DM) | 295.00–1340.11 | 6823–6711, 6245–6133, 4509–4397, 4278–4050 | 7 | 0.92 | 81.00 | 13.3 | $3.75^{-12}$ | 0.83 | 116.80 | 19.1 | $7.20^{-2}$ | 2.8 |
| Carotenoids (mg/kg DM) | 0.68–7.14 | 4586–4494 | 3 | 0.89 | 0.42 | 21.2 | $8.88^{-16}$ | 0.86 | 0.51 | 25.8 | $4.33^{-2}$ | 2.4 |
| Tocols (mg/kg DM) | 103.22–378.22 | 6245–6017 | 3 | 0.90 | 18.20 | 6.4 | $1.71^{-13}$ | 0.86 | 23.80 | 8.4 | $9.72^{-2}$ | 2.5 |
| FRAP (mmol TE/kg DM) | 3.90–30.90 | 9000–3800 | 2 | 0.80 | 2.56 | 14.2 | $-3.55^{-12}$ | 0.76 | 2.89 | 16.0 | $1.77^{-2}$ | 2.4 |

CAL, calibration; CV, cross-validation; $R^2$, coefficient of determination; $RMSE_{CAL}$, root mean square error in calibration; $RMSPE_{CAL}$, root mean square percentage error in calibration; $RMSE_{CV}$, root mean error in cross-validation; $RMSPE_{CV}$, root mean square percentage error in cross-validation; RPD, ratio of performance to deviation. Carotenoids and tocols data are from a previous study [6].

Even if the numerosity of the analysed samples is not substantial for model validation in prediction, its performance in cross-validation was evaluated in terms of retrieval, in respect to conventional method results. The average retrieval of the regression models was 100.0, 100.7, 105.1, 99.6 and 92.4% for total phenolics, flavonoids, carotenoids, tocols and antioxidant capacity, respectively. This means that on average the PLS models retrieve the compounds content as the reference method for total phenolics, flavonoids and tocols. The PLS model developed for carotenoids tends to overestimate the presence of compounds in the sample; instead, the model to predict FRAP– on average-underestimates the antioxidant capacity, in respect to the one measured by the colorimetric assay.

## 4. Conclusions

Even after a drastic debittering process that included boiling and recurrent washing, the Andean lupins maintained good free phenolic compounds concentrations, ranging from 340.81 to 1393.32 mg/kg DM. The flavonoids represented about 95% of the total free phenolics. The *L. albus* and *L. angustifolius* used as controls had different phenols composition and content, while the *L. luteus* had a similar profile but inferior values. Accordingly, the antioxidant capacity of the *L. mutabilis* ecotypes exceeded that of the controls and was largely correlated to flavonoids concentration. Five ecotypes (H6 INIA BP, Chacas, Moteado beige, Huánuco 2 and Lircay) showed excellent phenolics concentration and antioxidant capacity. Our results demonstrate that after debittering, Andean lupin

seeds are still rich in phenolic compounds with high antioxidant value, and as such are very promising for the preparation of high, nutritional value food products for human consumption. Additionally, the FT-NIR results demonstrated the relationship between the free phenolic compounds and the spectral bands, thus paving the way for a fast, reliable and non-destructive approach to lupin seed characterisation for phenolics content and antioxidant capacity. To this end, an increase of the lupin seed dataset would be critical to strengthen the prediction reliability of the regression models.

**Supplementary Materials:** The following supporting information can be downloaded at: https://www.mdpi.com/article/10.3390/pr10081637/s1, Figure S1: Areas of origin of the 33 *Lupinus mutabilis* ecotypes analysed; Figure S2: HPLC chromatogram of the free phenolics in one *Lupinus mutabilis* ecotype; Table S1: *Lupinus mutabilis* ecotypes and other *Lupinus* species tested, and their area of origin; Table S2: Free phenolics content (mean ± sd; mg/kg DM) in debittered seeds of *Lupinus ecotypes.*

**Author Contributions:** L.E.: investigation, formal analysis, methodology, software, writing—original draft, writing—review and editing; S.G.: investigation, methodology, writing—original draft, writing—review and editing; L.B.-B.: investigation, writing—review and editing; P.G.-P.: investigation, writing—review and editing; F.C.: investigation, writing—review and editing; A.H.: conceptualization, data curation, investigation, methodology, supervision, writing—original draft, writing—review and editing; A.B.: conceptualization, investigation, writing—original draft, writing—review and editing. All authors have read and agreed to the published version of the manuscript.

**Funding:** Partial research funding was provided by PNIA (Project 22-2015-INIA/UPMSI/IE).

**Data Availability Statement:** The data presented in this study are available on request from the corresponding author.

**Acknowledgments:** We thank Laura Steiner for her technical assistance.

**Conflicts of Interest:** The authors declare no conflict of interest.

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
