# Peer review of "Free Phenolic Compounds, Antioxidant Capacity and FT-NIR Survey of Debittered Lupinus mutabilis Seeds"

_processes, doi:10.3390/pr10081637_

Round 1

Reviewer 1 Report

This original article addresses a subject of interest: the assessment of free phenolic compounds, antioxidant capacity and FT-NIR spectra of lupin seed flours.

It deserves to be published in Processes, however, there are several points in which improvements should be made to strengthen the manuscript. 

1. Abstract does not mention carotenoids assay

2. Rows 90-91 - Please explain what sensorial analysis consists of.

3. Materials and methods does not describe methods used for tocols and carotenoids assay.

4. Rows 108-113 - HPLC method is not presented in full details (mobile phase, gradient elution, detection wavelength). Please detail or you may choose to cite your previously published paper, instead (Tocopherols, carotenoids and phenolics changes during Andean lupin (Lupinus mutabilis Sweet) seeds processing - ScienceDirect). A representative chromatogram is welcomed (inside the article or inserted in supplementary material)

5. Rows 119-124 - Please explain how detection limits were calculated (I found some differences between values presented in this manuscript and the ones mentioned in your previously published paper - please see observation 4)

6. Section 2.2.4 - Please describe sample holder and detector

7. My major concern is that the number of calibration samples is only 33. It is good that you mentioned this aspect as a limitation of the study (in Conclusions section)

8. Table 4 - Please present RMSEcal and RMSEcv in terms of %. I also recommend to state Bias

9. A comparison of results obtained by conventional methods and ones predicted by NIR chemometric methods (in terms of retrieval - for example 9 by means of conventional method and 8.8 predicted value, retrieval is 97.78%) would improve the value of your manuscript.

Author Response

We thank the Reviewer for the useful comments. We believe that they contributed to improve the manuscript.

Reviewer 1

This original article addresses a subject of interest: the assessment of free phenolic compounds, antioxidant capacity and FT-NIR spectra of lupin seed flours.

It deserves to be published in Processes, however, there are several points in which improvements should be made to strengthen the manuscript.

  1. Abstract does not mention carotenoids assay

Answer: Carotenoids of the very same samples were analysed in a recent paper (Briceño Berru, L.; Glorio-Paulet, P.; Basso, C.; Scarafoni, A.; Camarena, F.; Hidalgo, A.; Brandolini, A. Chemical composition, tocopherol and carotenoid content of seeds from different Andean lupin (Lupinus mutabilis) ecotypes. Plant Food Hum. Nutr. 2021, 76, 98–104. http://dx.doi.org/10.1007/s11130-021-00880-0). We have added the lines 78-80, 175, 335, 416-417 to clarify this work follows up our previous paper.

  1. Rows 90-91 - Please explain what sensorial analysis consists of.

Answer: We did not perform a real sensory analysis we just performed a tasting control to confirm the efficiency of the debittering process. On lines 102-103 we have replaced “sensorial analysis” with “tasting the grains for the absence of bitterness.”

  1. Materials and methods does not describe methods used for tocols and carotenoids assay.

Answer: Tocols and carotenoids of the very same samples were analysed in a recent paper (Briceño Berru, L.; Glorio-Paulet, P.; Basso, C.; Scarafoni, A.; Camarena, F.; Hidalgo, A.; Brandolini, A. Chemical composition, tocopherol and carotenoid content of seeds from different Andean lupin (Lupinus mutabilis) ecotypes. Plant Food Hum. Nutr. 2021, 76, 98–104.). In our FT-NIR preliminary study we considered these previous results. We have improved the description in different parts of the manuscripts (aims, lines 78-80 and materials and methods, line 175, line 335, lines 416-417).

  1. Rows 108-113 - HPLC method is not presented in full details (mobile phase, gradient elution, detection wavelength). Please detail or you may choose to cite your previously published paper, instead (Tocopherols, carotenoids and phenolics changes during Andean lupin (Lupinus mutabilis Sweet) seeds processing - ScienceDirect). A representative chromatogram is welcomed (inside the article or inserted in supplementary material)

Answer: The previously published paper has been cited and a representative chromatogram has been added as supplementary material (Figure S2).

  1. Rows 119-124 - Please explain how detection limits were calculated (I found some differences between values presented in this manuscript and the ones mentioned in your previously published paper - please see observation 4)

Answer: We have added a reference explaining the calculation of detection limit (21. Miller, J.C.; Miller, J.N. Statistics for Analytical Chemistry, 2nd ed.; Ellis Horwood Publishers: Chichester, UK, 1998). The Reviewer is right, there were some incongruences with reference to the previous manuscript. We decided to report the exact lowest concentration range used for the calibration curve instead of 0; the different detection limits reported for diosmin and cinnamic acid were typewriting mistakes, the difference observed for apigenin was a consequence of a different range of concentration, we constructed a new calibration curve considering a broader range to cover the higher variability of the values. We have carefully corrected the text in lines 131-139.

  1. Section 2.2.4 - Please describe sample holder and detector

Answer: The required descriptions have been added in the section 2.2.4.

  1. My major concern is that the number of calibration samples is only 33. It is good that you mentioned this aspect as a limitation of the study (in Conclusions section)

Answer: As stated by the Reviewer, we are aware that the number of samples is quite low (38 samples) to develop a PLS model and we agree we would need to increase the numbers and to validate the prediction ability of the model with an adequate prediction step.

Similar works, cited in the manuscript, developed regression models with 21 samples (ref #46) or 49 samples (ref #42). Even Tomar et al [41], who developed a complete regression approach, i.e. including calibration, cross-validation and prediction, built the model with only 53 samples. This is to say that there are specific cases in which a statistically sound number of samples is difficult to reach. In any case, we have highlighted also in the discussion the scarce number of samples, as already done in the conclusion section. Furthermore, this is just a part of the work, which is primarily intended to improve the characterization of lupins by determining the free phenolic compounds and antioxidant capacity of the water-debittered seeds.

  1. Table 4 - Please present RMSEcal and RMSEcv in terms of %. I also recommend to state Bias

Answer: We would like to thank the Reviewer for the precious observation. We have added the bias as well as the RMSPE for both calibration and cross-validation. Anyway, we would prefer to maintain in the text, and in the table, the error express in the compounds unit.

  1. A comparison of results obtained by conventional methods and ones predicted by NIR chemometric methods (in terms of retrieval - for example 9 by means of conventional method and 8.8 predicted value, retrieval is 97.78%) would improve the value of your manuscript.

Answer: A comparison of the results obtained by the conventional methods and the NIR regression models has been added at the end of the results section.

Reviewer 2 Report

This manuscript reports the composition in phenolic compounds, the antioxidant capacity, and the FT-NIR spectra of debittered seed samples of 33 Andean ecotypes of L. mutabilis and 5 control varieties belonging to L. luteus, L. angustifolius and L. albus.

The manuscript is well written and structured, but lacks some novelty compared to other studies found in the literature, in part due to the implemented experimental design and the followed methodological approaches.

Major comments/observations:

-        The analyses should have been done before and after the debittering process. In this way, it would have been possible to know the original chemical composition of the samples and the suitability of the studied debittering process. Without the initial characterization, the impact of the process is unknown.

-        The 33 Lupinus mutabilis ecotypes from the Peruvian Andes were characterized. As reference, 2 L. albus, 1 Lupinus angustifolius and 1 Lupinus luteus from Italy and 1 Lupinus albus from Perú were considered. However, it is not clear why these samples were used as a reference for comparing results.

-        Since this study intends to compare ecotypes, it will be important to provide more data on the origin of the ecotypes described in Supplementary Table 1. The observed chemical differences may be due to the ecotype but also to the agricultural practices used in their production and the soil and climate conditions of the different production sites. Therefore, the authors should be able to prove that the chemical differences are due to the ecotype and not to agricultural practices or other variation factors. Furthermore, it will be relevant to indicate whether the samples of each ecotype are from one or more production years, as well as from one or more agricultural farms.

-        Authors should also make sure that the term "ecotype" can be used to describe the studied lupin accessions.

-        Phenolic compounds were identified “by the congruence of retention times and UV/VIS spectra with those of pure authentic standards”. The identity of the detected phenolic compounds should have been confirmed with mass spectrometry.

Minor observations/observations:

-        In line 22 and other sections, the number of samples must be indicated in the same way, i.e., 33, 5… or thirty-three, five…

-        Lines 69 and 70. It is referred that “The widespread industrial utilisation of the Andean lupin is hampered by the limited number of studies analysing its composition and technological characteristics.” Is this or only this the main reason that limits the use of these ecotypes at an industrial level?

-        Line 74. It is mentioned that it was intended “to identify the most suitable ecotypes for developing high nutritional value food products”. However, this study focused mainly on phenolic compounds and antioxidant properties, but the analysis of other nutritional quality attributes (e.g., contents of protein, minerals, dietary fibre, or antinutrients, among others) was not evaluated. Therefore, the sentence should be reformulated.

-        Line 115. It is mentioned that “Thirty-three standards were injected”, but only seven calibration curves were constructed. This information can be clarified for a better understanding.

-        Line 157. “traits analysed in this work and in previous related research [6]”. It will be important to mention whether the plant material analysed in both studies was exactly the same.

-        Scientific names should be written in italics throughout the manuscript.

-        Statistical differences can be shown in Figures 1 and 2, for example by using different letters on top of the bars of the graphs.

-        It was concluded that “Even after a drastic debittering process that included boiling and recurrent washing, the Andean lupins maintained good free phenolic compounds concentrations”. However, since the contents of compounds were not analysed in the samples before debittering, it will not be possible to know whether the process was associated with a significant loss of compounds or not.

Author Response

We thank the Reviewer for the useful comments. We believe that they contributed to improve the manuscript.

Reviewer 2

This manuscript reports the composition in phenolic compounds, the antioxidant capacity, and the FT-NIR spectra of debittered seed samples of 33 Andean ecotypes of L. mutabilis and 5 control varieties belonging to L. luteusL. angustifolius and L. albus.

The manuscript is well written and structured, but lacks some novelty compared to other studies found in the literature, in part due to the implemented experimental design and the followed methodological approaches.

Answer: Actually this manuscript, along with its companion paper (Briceño Berru, L.; Glorio-Paulet, P.; Basso, C.; Scarafoni, A.; Camarena, F.; Hidalgo, A.; Brandolini, A. Chemical composi-tion, tocopherol and carotenoid content of seeds from different Andean lupin (Lupinus mutabilis) ecotypes. Plant Food Hum. Nutr. 2021, 76, 98–104. http://dx.doi.org/10.1007/s11130-021-00880-0), represents the first comprehensive approach to characterise the antioxidant properties of a broad spectrum of Peruvian L. mutabilis germplasm from different eco-geographical regions using HPLC and FT-NIR.

Major comments/observations:

- The analyses should have been done before and after the debittering process. In this way, it would have been possible to know the original chemical composition of the samples and the suitability of the studied debittering process. Without the initial characterization, the impact of the process is unknown.

Answer: The impact of the debittering process on lupin antioxidants is not the aim of the present work. Debittering effect on Andean lupin seeds antioxidant properties was already deeply investigated in former papers (Brandolini, A.; Glorio-Paulet, P.; Estivi, L.; Locatelli, N.; Córdova-Ramos, J. S.; Hidalgo, A. Tocopherols, carotenoids and phenolics changes during Andean lupin (Lupinus mutabilis Sweet) seeds processing. J. Food Compos. Anal. 2022, 106, 104335. https://doi.org/10.1016/j.jfca.2021.104335; Córdova-Ramos, J.S.; Glorio-Paulet, P.; Camarena, F.; Brandolini, A.; Hidalgo, A. Andean lupin (Lupinus mutabilis Sweet): processing effects on chemical composition, heat damage and in vitro protein digestibility. Cereal Chem. 2020, 97, 827–835. https://doi.org/10.1002/cche.10303; Córdova-Ramos, J.S.; Glorio-Paulet, P.; Hidalgo, A.; Camarena, F. Effect of technological process on antioxidant capacity and total phenolic content of Andean lupine (Lupinus mutabilis Sweet). Sci. Agropecu. 2020, 11, 157–165. https://doi.org/10.17268/sci.agropecu.2020.02.02.).

Anyway, it is of low interest for food consumption the investigation of phenolics in the bitter seeds. As already reported, the bitter seeds are rich in phenolics that are removed together with the alkaloids during the washing steps of the debittering process.

- The 33 Lupinus mutabilis ecotypes from the Peruvian Andes were characterized. As reference, 2 L. albus, 1 Lupinus angustifolius and 1 Lupinus luteus from Italy and 1 Lupinus albus from Perú were considered. However, it is not clear why these samples were used as a reference for comparing results.

Answer: L. albus, L. luteus and L. angustifolius are the three species commonly cropped all around the world. Therefore, to fully appreciate the characteristics of the Peruvian ecotypes of L. mutabilis we decided to include some varieties of the above-mentioned species as controls.

- Since this study intends to compare ecotypes, it will be important to provide more data on the origin of the ecotypes described in Supplementary Table 1. The observed chemical differences may be due to the ecotype but also to the agricultural practices used in their production and the soil and climate conditions of the different production sites. Therefore, the authors should be able to prove that the chemical differences are due to the ecotype and not to agricultural practices or other variation factors. Furthermore, it will be relevant to indicate whether the samples of each ecotype are from one or more production years, as well as from one or more agricultural farms.

Answer: The seeds of the 33 ecotypes were originally collected in different eco-geographical areas of Peru and stored in the facilities of the Universidad Nacional Agraria La Molina, Lima, Peru. In 2018 all the ecotypes were reproduced in the Lima field of said university, following the same agricultural practices, therefore no differences between the Peruvian samples (and the Dulce 7 L. albus) can be attributed to agricultural practices, soil and climate conditions. These effects are theoretically possible for the other controls, cropped in Italy, however their performances fall within the known and published range of each species. We modified the phrase in line 90 to “… were originally collected …”

- Authors should also make sure that the term "ecotype" can be used to describe the studied lupin accessions.

Answer: An ecotype is “a distinct form or race of a plant or animal species occupying a particular habitat”. Our Peruvian L. mutabilis samples fit perfectly into this definition, as they are not varieties, and they evolved and were cropped in different and particular habitats.

- Phenolic compounds were identified “by the congruence of retention times and UV/VIS spectra with those of pure authentic standards”. The identity of the detected phenolic compounds should have been confirmed with mass spectrometry.

Answer: At present we are not able to comply with this request because we do not have the indicated instrumentation.  Therefore, to perform a further identification of the derivative peaks we should involve other researchers. If an exact match for the spectrum can be found, but the RT differs, then it is likely that the unknown compound differs from the reference only in the number or nature of sugars attached to the glycosylated site, or in the nature of a substituent, or presence/absence of an aliphatic acyl group on a sugar (from: Campos, M.G., Markham, K.R. 2007. Structure information from HPLC and on-line measured absorption spectra: flavones, flavonols and phenolic acids. Page 13. DOI:http://dx.doi.org/10.14195/978-989-26-0480-0), hence several Authors adopt the derivative or equivalent approach (e.g. Duenas et al., 2009; Zalewski et al., 2020; cited in our article). Nevertheless, we will strive to include your suggestion in our future researches.

Minor observations/observations:

- In line 22 and other sections, the number of samples must be indicated in the same way, i.e., 33, 5… or thirty-three, five…

Answer: Done

- Lines 69 and 70. It is referred that “The widespread industrial utilisation of the Andean lupin is hampered by the limited number of studies analysing its composition and technological characteristics.” Is this or only this the main reason that limits the use of these ecotypes at an industrial level?

Answer: This is the main reason; of course, a further cause is the lack of improved varieties with adaptation to European, North American and Australian environments, but to breed such varieties the first step is a good understanding of the properties of the existing germplasm. We have modified the phrase in Lines 76-77 to include this motive: “The widespread industrial utilisation of the Andean lupin is hampered by the limited number of studies analysing its composition and technological characteristics as well as to the lack of improved varieties suitable for cultivation in the main lupin cropping areas.

- Line 74. It is mentioned that it was intended “to identify the most suitable ecotypes for developing high nutritional value food products”. However, this study focused mainly on phenolic compounds and antioxidant properties, but the analysis of other nutritional quality attributes (e.g., contents of protein, minerals, dietary fibre, or antinutrients, among others) was not evaluated. Therefore, the sentence should be reformulated.

Answer: Several other nutritional quality attributes (chemical composition, tocols and carotenoids content) were studied in a recent companion article (Briceño Berru, L.; Glorio-Paulet, P.; Basso, C.; Scarafoni, A.; Camarena, F.; Hidalgo, A.; Brandolini, A. Chemical composi-tion, tocopherol and carotenoid content of seeds from different Andean lupin (Lupinus mutabilis) ecotypes. Plant Food Hum. Nutr. 2021, 76, 98–104. http://dx.doi.org/10.1007/s11130-021-00880-0). This information has been added to the introduction (lines 78-80).

- Line 115. It is mentioned that “Thirty-three standards were injected”, but only seven calibration curves were constructed. This information can be clarified for a better understanding.

Answer: We injected thirty-three standards and constructed their calibration curves since the HPLC system is used to analyse different sample materials. However, this information is useless and “Thirty-three standards were injected” has been removed.

- Line 157. “traits analysed in this work and in previous related research [6]”. It will be important to mention whether the plant material analysed in both studies was exactly the same.

Answer: Yes, the plant materials analysed in both articles were exactly the same. This aspect has been emphasized by modifying the text in lines 81-82, 175, 335, 416-417.

- Scientific names should be written in italics throughout the manuscript.

Answer: Done

- Statistical differences can be shown in Figures 1 and 2, for example by using different letters on top of the bars of the graphs.

Answer: Considering that we were analysing 33 Andean lupins and 5 controls (i.e. 38 samples), it is probably impossible and anyway highly unsuitable, cumbersome and illegible to use different letters to highlight statistical differences. In the Figures the standard deviations are depicted by the error bars. Anyway, the comments in the text are based on the LSD test performed after ANOVA.

- It was concluded that “Even after a drastic debittering process that included boiling and recurrent washing, the Andean lupins maintained good free phenolic compounds concentrations”. However, since the contents of compounds were not analysed in the samples before debittering, it will not be possible to know whether the process was associated with a significant loss of compounds or not.

Answer: The impact of the debittering process on Andean lupin seeds antioxidant properties was already deeply investigated in former papers (Brandolini et al. J. Food Compos. Anal. 2022, 106, 104335. https://doi.org/10.1016/j.jfca.2021.104335; Córdova-Ramos et al. Cereal Chem. 2020, 97, 827–835. https://doi.org/10.1002/cche.10303; Córdova-Ramos et al. Sci. Agropecu. 2020, 11, 157–165. https://doi.org/10.17268/sci.agropecu.2020.02.02.). As already reported in the above-mentioned articles, the bitter seeds are rich in phenolics that are removed together with the alkaloids during the washing steps of the debittering process. Anyway, the investigation of phenolics in the bitter seeds is of little interest human consumption.

Round 2

Reviewer 2 Report

Most of the Reviewer's comments and suggestions were considered during the preparation of the revised version of the manuscript. However, not all rebuttals given by the authors appeared to be sufficiently valid or credible.

Line 102 and 103. "tasting the grains for the absence of bitterness". It would be interesting to indicate how these tastings were conducted and by whom.

Author Response

We thank the Reviewer for the comment.

The debittering method applied to all the samples was always the same. The tasting of the seeds, performed by the operators, was simply an additional (and not necessary) verification of the efficiency of the method, traditionally used in research and industry. For example, it was used as a control procedure in a recently published paper (Estivi, L.; Buratti, S.; Fusi, D.; Benedetti, S.; Rodríguez, G.; Brandolini, A.; Hidalgo, A. Alkaloid content and taste profile as-sessed by electronic tongue of Lupinus albus seeds debittered by different methods. J. Food Comp. Anal. 2022, 104810. https://doi.org/10.1016/j.jfca.2022.104810) upholding the efficiency of the debittering process through the residual alkaloid content by titration (von Baer, D., Reimerdes, E.H., Feldheim, W., 1979. Methoden zur Bestimmung der Chinolizidinalkaloide in Lupinus mutabilis. Z. für Lebensm.-Unters. und Forsch. 169, 27–31. https://doi.org/10.1007/BF01353410) and the electronic tongue analysis.

The text was changed to “This method efficiently removes alkaloids and bitterness, as demonstrated by Estivi et al. [19] using colorimetric titration and electronic tongue.”
